# A Rare Case of Dedifferentiated Liposarcoma with Osteosarcomatous Differentiation-Diagnostic and Therapeutic Challenges

**DOI:** 10.3390/diseases12010006

**Published:** 2023-12-25

**Authors:** Patrycja Sosnowska-Sienkiewicz, Przemysław Mańkowski, Honorata Stadnik, Agata Dłubak, Anna Czekała, Marek Karczewski

**Affiliations:** 1Department of Pediatric Surgery, Traumatology and Urology, Poznan University of Medical Sciences, Szpitalna Street 27/33, 60-572 Poznan, Poland; mankowskip@ump.edu.pl; 2Department of General and Transplant Surgery, Poznan University of Medical Sciences, 60-355 Poznan, Poland; honorata.stadnik@usk.poznan.pl (H.S.); agata.dlubak@usk.poznan.pl (A.D.); mkar@ump.edu.pl (M.K.); 3Department of Clinical Pathology, Poznan University of Medical Sciences, 60-355 Poznan, Poland; annamariaczekala@gmail.com

**Keywords:** dedifferentiated liposarcoma, laparoscopy, MDM2, osteogenic differentiation, sarcoma

## Abstract

Introduction: Liposarcomas are the most common of all sarcomas. A well-differentiated liposarcoma can transform into a dedifferentiated liposarcoma with myogenic, osteo- or chondrosarcomatous heterologous differentiation. Genomic amplification of *MDM2* gene is then characteristic. Treatment usually involves surgical resection to radically remove the tumor. Other treatments such as chemotherapy and radiotherapy may also be used. Case report: A 60-year-old patient was admitted to the hospital for surgical treatment of a left renal mass. The true location of the tumor was discovered only intraoperatively. The lesion was completely removed laparoscopically with preservation of the capsule. Genomic amplification of *MDM2* gene was confirmed. One and a half years after surgery, despite the removal of the tumor without the surrounding margin of healthy tissue, the patient remains without recurrence. Conclusion: Dedifferentiated liposarcoma with osteosarcomatous differentiation is a sporadic case and may occur in various locations of the retroperitoneal space, also mimicking a renal tumor. The laparoscopic technique is a safe surgical treatment for tumors of unclear origin. Removal of dedifferentiated liposarcoma with osteosarcomatous differentiation tumor with preservation of the lesion capsule without maintaining a margin of healthy tissue also allows for long-term cure. Precise immunohistochemical and molecular studies may have an impact on the effectiveness of further treatment and the prognosis of the patient. A patient after surgical treatment of liposarcoma requires constant outpatient follow-up for the reason of the high risk of local and distant recurrence.

## 1. Introduction

Soft tissue sarcomas are heterogeneous lesions derived from mesenchymal cells, mainly from adipose tissue, nerves, vessels, and connective tissue, and there are more than 100 subtypes [1]. They are a relatively rare and diagnostically challenging group of cancers. Proper diagnosis is important for appropriate treatment and prognosis [2]. Classification of neoplasms, especially soft tissues, is increasingly based on molecular characterization of tumor types. Understanding the molecular genetics of neoplasms improves the diagnostic accuracy of those tumors that have been difficult to classify on the basis of morphology alone or those that have overlapping morphological features of different diagnoses [2]. In many major hospitals, molecular testing of soft tissue tumors is a routine part of diagnostics [2,3].

Liposarcomas (LPSs) are the most common, accounting for approximately 20% of all sarcomas [4]. The most frequent sites of involvement of LPS are the retroperitoneum, the deep soft tissues of extremities, and the spermatic cord. Paravertebral involvement is also described [3]. LPSs of the gastrointestinal tract occur very rarely and can be found in the stomach, small intestine, and colon. Localization is also possible in the rectum [5]. LPSs are connected with certain mutations in adipocytes, causing the cells to begin to divide indefinitely and form a tumor mass. The causes of these mutations are unknown. All that is known is that a higher incidence of soft tissue sarcomas is associated with some rather rare genetic syndromes (e.g., neurofibromatosis types 1 and 2, Li and Fraumeni syndrome, Gardner syndrome, and Werner syndrome), in immunocompromised patients and after exposure to ionizing radiation and certain chemical agents (e.g., herbicides, pesticides, polyvinyl chloride, and dioxins) [6]. A well-differentiated liposarcoma (WDL) may evolve into a dedifferentiated liposarcoma (DDL) with pleomorphic histopathology. This process is often observed in the retroperitoneal space and is associated with an increased risk of metastasis and local recurrence [7]. In a small percentage of cases (5–10%), myogenic, osteo- or chondrosarcomatous heterologous differentiation is present [2,8]. Myogenic differentiation is the most common type, and the impact of myogenic differentiation on prognosis has been studied. Bone tissue formation has been reported in 0.6–29% of cases in large studies of DDL, and bone formation was most commonly described as “osteosarcoma” or “metaplastic osteogenesis” [5]. In contrast to myogenic differentiation, the clinical significance of osteogenic differentiation in DDL has not been systematically studied [9]. Other mesenchymal lineages such as chondrosarcoma and angiosarcoma have also been reported rare [10].

The WDL/DDL group is characterized by gene amplification in the 12q chromosomal region 13–15, especially the *MDM2* (mouse double minute 2 homolog) gene [7].

Liposarcoma occurs most commonly in adult patients, but can occur at any age. Treatment usually involves surgical resection to radically remove the tumor. Other treatments, such as radiation therapy, may also be used. For unresectable, advanced, or metastatic disease, the efficacy of chemotherapy may vary by subtype [11].

Liposarcoma has a relatively good prognosis. Complete cure of liposarcoma depends on the neoplasm subtype, location, and stage at the time of diagnosis (including tumor size, presence of metastases). The best prognosis is for the well-differentiated and mucinous subtype. A worse prognosis is seen with the diagnosis of undifferentiated and multiforme liposarcoma, which are more likely to cause distant metastases [12].

The aim of this publication was to present a rare clinical case of dedifferentiated liposarcoma with osteosarcomatous differentiation, initially interpreted as a renal tumor, which was successfully removed using the laparoscopic technique.

## 2. Case Report

A 60-year-old female patient was admitted to the hospital in December 2022 for surgical treatment of a tumor in the left renal region. The patient reported periodic abdominal pain, general weakness, and no other complaints for about half a year. A follow-up abdominal ultrasound (USG) revealed a tumor in the left renal region. Computed tomography (CT) of the abdomen, pelvis, and chest was added to the diagnostic work-up. A CT scan revealed a 56 mm heterogeneous mass in the region of the left renal hilum with an equivocal exit point (Figure 1).

Other examinations showed no abnormalities in the patient. No pathological lesions were observed in the lungs. No pathologically changed lymph nodes were visualized. Because of the typical clinical presentation, the patient was qualified for laparoscopic renal tumor removal with preservation or complete removal of the kidney, depending on the intraoperative findings. Intraoperatively, a tumor was visualized that was not directly related to the kidney. It was located near the renal hilum and was closely related to the renal vessels. The lesion could be removed radically with a minimally invasive technique during laparoscopic evaluation. The location of the tumor made resection with a margin of healthy tissue impossible. Achieving a resection margin would require removal of the kidney. The lesion was completely dissected macroscopically from the renal vessels and surrounding tissues with preservation of the capsule (Figure 2 and Figure 3).

The mass was removed with all the rules of oncological cleanliness in an endoscopic bag. The surgical treatment was uneventful. The patient was discharged home after 5 days of observation.

As a result of macroscopic histopathologic examination, the surgical material was described as a solid, polycyclic, cream-colored tumor, partly with a cyst. The cystic part was filled with yellow fluid. In addition, hemorrhagic changes and massive calcifications were observed within the tumor. Histopathologic examination described cytologic atypia as high grade with mitotic activity: 10/1734 mm^2^ and no signs of necrosis. Immunohistochemistry (IHC) results were: MDM2 (+/−) focal, CDK4 (+/−), desmin (−), myogenin (−), caldesmon (−). Numerous multinucleated giant cells with osteoclast morphology were mainly located around the hemorrhage and ossification. Amplification of the *MDM2* gene was detected via fluorescence in situ hybridization (FISH). The final diagnosis was described as dedifferentiated liposarcoma FNCLCC (Fédération Nationale de Centres de Lutte Contre le Cancer), G2 (Grade 2) with osteosarcomatous differentiation, pT2 N0 (Figure 4, Figure 5 and Figure 6) [13].

The histopathological result showed no margin of healthy tissue surrounding the removed tumor, but its capsule was preserved (Appendix A).

The patient continues to be followed on an outpatient basis. Follow-up examinations (CT of the chest, abdomen, and pelvis) performed 1.5 years after surgery showed no evidence of local or distant recurrence. The patient is in good general condition without any complaints.

## 3. Discussion

Dedifferentiated liposarcoma with osteosarcomatous differentiation is a rare and aggressive form of cancer that involves the transformation of liposarcoma into a tumor with additional osteosarcoma-like features [1].

Liposarcomas are generally classified into several subtypes based on their histologic features and biological behavior. Dedifferentiated liposarcoma is characterized by both well-differentiated liposarcoma and a non-lipogenic sarcomatous component. This non-lipogenic component may exhibit various differentiation patterns, including osteosarcomatous differentiation [14]. Such a rare type of dedifferentiated liposarcoma was diagnosed in our patient. The location of the tumor on CT within the kidney did not predict such a diagnosis at all.

Due to their retroperitoneal location, liposarcoma tumors can often reach enormous sizes without causing any signs or symptoms. The tumor eventually reaches an average size of 15 cm when symptoms, often nonspecific to the abdominal cavity, appear. These include a feeling of fullness in the abdomen, early satiety with associated malnutrition, and a palpable mass on physical examination of the abdomen [1]. The patient who underwent surgery at our center had no other symptoms other than periodic abdominal pain. Such symptoms may occur for many reasons, further delaying a proper diagnosis [15].

The exact cause of dedifferentiated liposarcoma with osteosarcomatous differentiation is not well understood. However, some risk factors, such as prior radiation therapy, genetic abnormalities, and certain hereditary conditions, have been associated with the development of liposarcomas in general [1,8]. Amplification of the *MDM2* gene is characteristic of well-differentiated and dedifferentiated liposarcomas [8]. *MDM2* gene amplification was confirmed in our patient.

The diagnosis of dedifferentiated liposarcoma with osteosarcomatous differentiation involves a combination of clinical evaluation, imaging studies, and histopathological examination of the tumor. Imaging techniques such as computed tomography (CT) and magnetic resonance imaging (MRI) can help visualize the extent of the tumor and detect any associated bony changes. Contrast-enhanced computed tomography (CT) is the most commonly used scan [1]. Abdominal ultrasound is often the first investigation. The diagnostic process was similar in our center, where the patient had an ultrasound and a CT at the beginning of the diagnostic process. Positron emission tomography (PET) does not have a routine role in the diagnosis of sarcoma [1].

Biopsy confirmation of the diagnosis is routinely recommended for retroperitoneal sarcoma to facilitate optimal perioperative management after discussion with the multidisciplinary team [1]. In our patient, a biopsy was not performed. This was because the tumor may have been defined as originating from the kidney, where biopsy is not standard.

Treatment of dedifferentiated liposarcoma with osteosarcomatous differentiation usually involves a multidisciplinary approach. Surgery is the primary treatment modality, with the goal of removing the tumor and all surrounding affected tissues, including bone if necessary. In some cases, limb-sparing surgery may be possible, while in more advanced cases, amputation may be required. Adjuvant therapies, such as radiation therapy and chemotherapy, may be recommended before or after surgery to reduce the risk of recurrence and improve outcomes. The specific treatment plan is determined based on factors such as tumor size, location, grade, and the individual patient’s overall health [1]. There is no standard systemic adjuvant treatment for retroperitoneal sarcoma. The histopathological examination determines it. Systemic therapy in the neoadjuvant setting may result in cytoreduction of the primary tumor and reduce the risk of distant metastases [16]. There is also considerable variability in the timing, delivery, and dose of radiotherapy. The indications for radiotherapy are to reduce the risk of recurrence when there is an increased risk of local recurrence, especially in patients with positive margins after surgery, high-grade tumors, and certain histopathological findings [17]. Given the significant variation in natural history, histopathologic subtype is an important factor influencing the risk of local recurrence. For example, myxosarcoma is highly infiltrative, especially in the subcutaneous tissue, and tends to have multiple local recurrences, although it is associated with a lower risk of metastasis [18]. Malignant Peripheral Nerve Sheath Tumor (MPNST) is also highly infiltrative and is similarly associated with not only an increased risk of local recurrence, but also a high risk of metastasis [19]. The same treatment strategy seems appropriate for pleomorphic liposarcoma, but with much lower expectations for a satisfactory outcome. An important goal for this variant of liposarcoma is to develop an effective systemic therapy [20].

Soft tissue sarcomas include liposarcoma and leiomyosarcoma, leiomyosarcoma, and undifferentiated pleomorphic sarcoma entities. The current first-line treatment in these cases is represented by anthracycline-based regimens, and second-line treatments may include trabectedin. Currently, the activity of trabectedin and its mechanism of action are not fully understood [21].

In our patient, it was possible to perform surgery by removing the tumor without disturbing its capsule. The tumor was carefully separated from the vessels and adjacent structures. By reason of the fact that obtaining the margin would require the removal of the renal vessels and the removal of the kidney, it was decided to preserve the organ. Due to the size and location of the lesion, no preoperative therapy was performed. Postoperative radiotherapy was also not chosen. After further consultation with the Rare Tumor Treatment Center outside of our clinical hospital, additional treatment was abandoned. Observation was indicated as the correct course. USG every 6 months and abdominal CT every year were recommended.

In clinical practice, it is also important to have a comprehensive clinical work-up for the identification of possible synchronous tumors that may have an impact on the patient’s treatment and prognosis [5].

The prognosis of dedifferentiated liposarcoma with osteosarcomatous differentiation tends to be worse than that of other liposarcoma subtypes. The aggressive nature of the tumor and its potential for metastasis contribute to the challenges of effectively managing this disease. The overall and disease-free survival rates depend on several factors [1,3]. In our case, the positive aspect is that the tumor was completely removed with the capsule. Outpatient follow-up with regular, appropriate diagnostics is critical [10].

The histopathologic subtype of the neoplasm may be difficult to determine. Precise immunohistochemical and molecular studies may be necessary. This may have an impact on the effectiveness of further treatment and the prognosis of the patient [3]. Detailed histopathologic, immunohistochemical and molecular diagnostics were important in our patient’s diagnosis. The final diagnosis also prompts increased post-operative care.

Considering the rarity of dedifferentiated liposarcoma with osteosarcomatous differentiation, it is essential that patients are treated in specialized sarcoma centers with experience in the management of these complex cases. Close monitoring, regular follow-up, and collaboration between oncologists, surgeons, and other health care professionals are critical to optimizing treatment outcomes and providing the best possible care to affected individuals [1]. Our patient required extreme caution and multidisciplinary discussion.

## 4. Conclusions

Dedifferentiated liposarcoma with osteosarcomatous differentiation is a sporadic case and may occur in various locations of the retroperitoneal space, also mimicking a renal tumor.

The laparoscopic technique is a safe surgical treatment for tumors of unclear origin.

Removal of dedifferentiated liposarcoma with osteosarcomatous differentiation tumor with preservation of the lesion capsule without maintaining a margin of healthy tissue also allows for long-term cure.

Precise immunohistochemical and molecular studies may have an impact on the effectiveness of further treatment and the prognosis of the patient.

A patient after surgical treatment of liposarcoma requires constant outpatient follow-up for the reason of the high risk of local and distant recurrence.

## Figures and Tables

**Figure 1 diseases-12-00006-f001:**
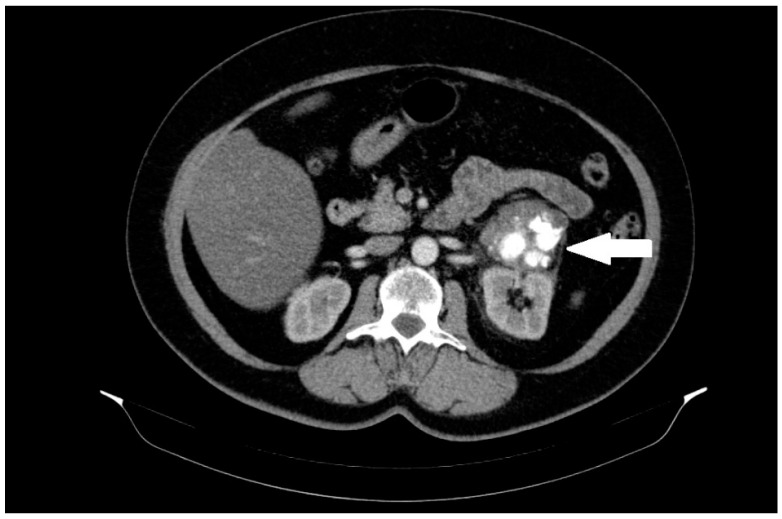
A heterogeneous solid mass, 56 mm in diameter, located between the left kidney, adrenal gland, and pancreas, with massive calcifications and an unclear exit point. The lesion is marked with an arrow.

**Figure 2 diseases-12-00006-f002:**
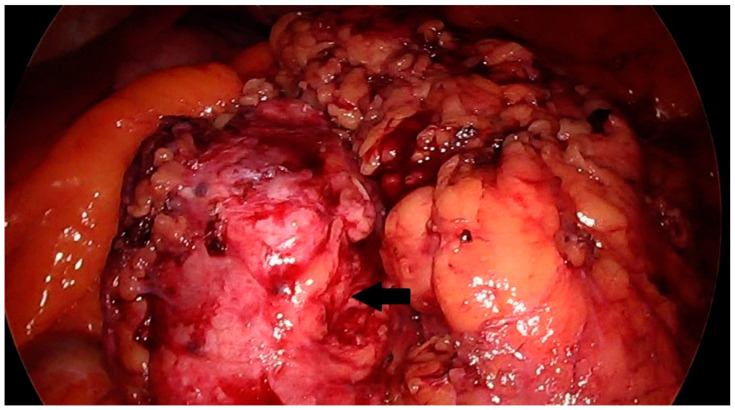
Solid mass in the region of the left kidney, intraoperative image. The prepared lesion is marked with an arrow.

**Figure 3 diseases-12-00006-f003:**
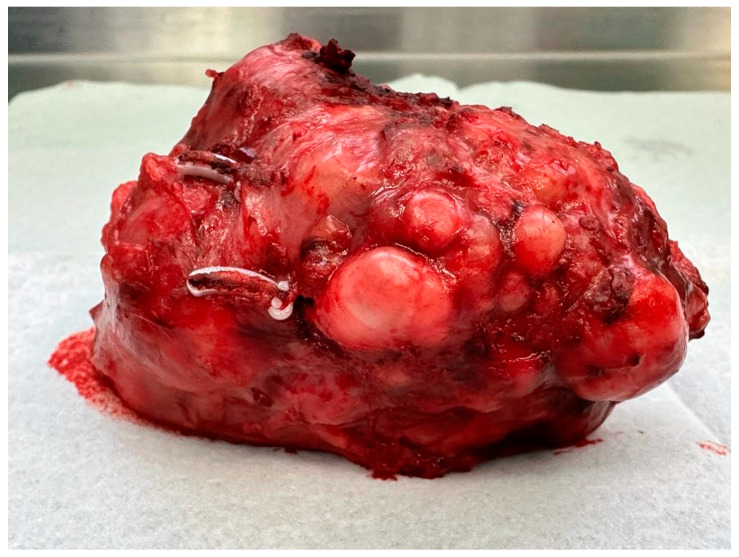
The tumor after resection.

**Figure 4 diseases-12-00006-f004:**
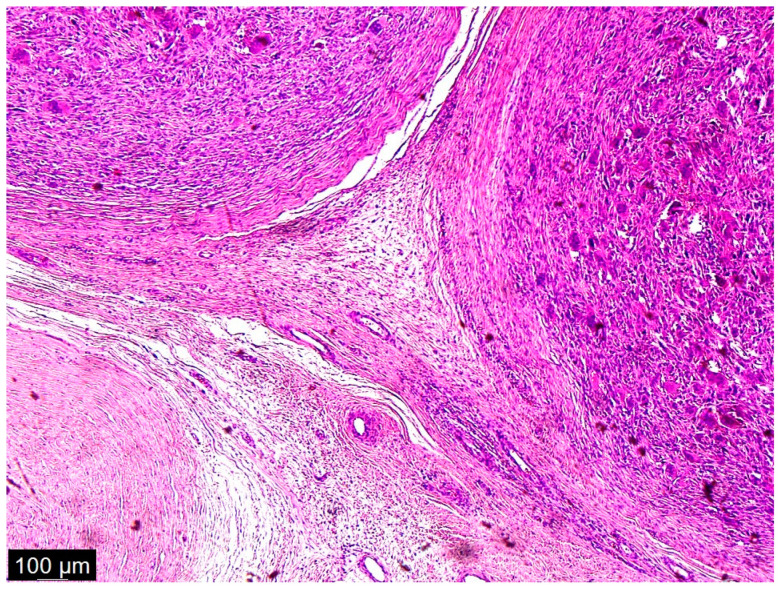
A low magnification of the tumor showed a nodular mass. The nodules were formed from multinucleated giant cells and mononuclear cells. Interspersed among these structures were spindle cells, some of which were pleomorphic and some of which were simply dark and hyperchromatic. This was not a typical case, because in the textbook definition of dedifferentiated liposarcoma, there are parts of well-differentiated liposarcoma and atypical squamous cells which we did not see in this case; there were just scattered spindle cells and a heterologous component.

**Figure 5 diseases-12-00006-f005:**
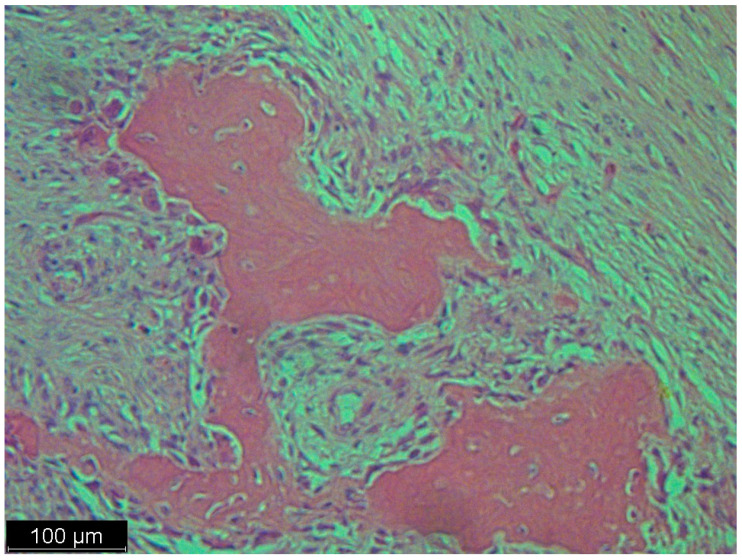
This picture looked like osteosarcoma, but it was not. It was a dedifferentiated liposarcoma with a heterologous component. In this case, the tumor produced osteoids with calcified trabeculae. The photo shows an osteoblastic rimming.

**Figure 6 diseases-12-00006-f006:**
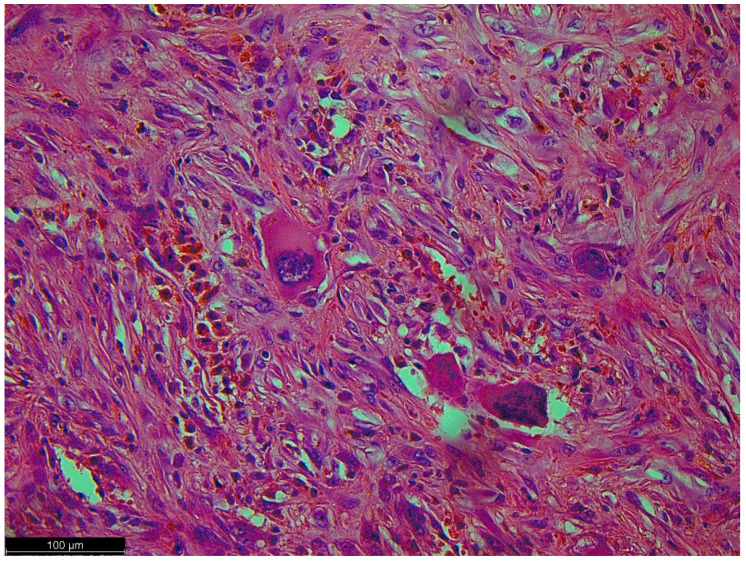
Pleomorphic cells with high-grade atypia; they did not show typical differentiation. There were no lipoblasts or remnants of well-differentiated liposarcoma (the precursor to this tumor) because a dedifferentiated component had overgrown it.

## Data Availability

The data presented in this study are available on request from the corresponding author.

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
