# Peer review of "A Rare Case of Dedifferentiated Liposarcoma with Osteosarcomatous Differentiation-Diagnostic and Therapeutic Challenges"

_diseases, 2023, doi:10.3390/diseases12010006_

Round 1

Reviewer 1 Report

Comments and Suggestions for Authors

Very interesting case report, both from a surgical and histological point of view

Comments on the Quality of English Language

I would only recommend to vary the begininng of some phrases since a lot of them start with "due to" and "there were".

Author Response

Dear Reviewer,

Thank you very much for your valuable remarks and constructive comments. We have addressed them point by point. Additionally, in the revised version of the manuscript, changed or added text sections have been marked in yellow.

We are submitting a revised manuscript entitled "A rare case of dedifferentiated liposarcoma with osteosarcomatous differentiation- diagnostic and therapeutic challenges" and we hope that the revised version will be found satisfactory for publication in Diseases.

Kind regards,

Patrycja Sosnowska-Sienkiewicz

  1. Very interesting case report, both from a surgical and histological point of view.

Re: Thank you sincerely for this comment. I am glad that the manuscript meets the conditions of a valuable publication.

  1. I would only recommend to vary the begininng of some phrases since a lot of them start with "due to" and "there were".

Re: Thank you very much for this comment. The phrases "due to" and "there were" have been replaced.

The entire publication was also rechecked and revised by a native speaker.

Thank you again for all your comments.

Kind regards,

Patrycja Sosnowska-Sienkiewicz

Reviewer 2 Report

Comments and Suggestions for Authors

The authors Sosnowska-Sienkiewicz and colleagues report an uncommon presentation of dedifferentiated liposarcoma with osteosarcomatous differentiation.

The manuscript is interesting and well described.

The manuscript would benefit from the followings:

1.       Latest WHO should be referenced: WHO Classification of Tumours. In Soft Tissue and Bone, 5th ed.; IARC Press: Lyon, France, 2020; Volume 3, p. 368, ISBN 978-92-832-4502-5.

2.       Diagnostic images of MDM2 amplification should be reported (i.e. FISH, PCR etc)

3.       The authors should include some reference from proper discussion

Synchronous occurrence of small cell lung cancer and primary rectal dedifferentiated liposarcoma with osteosarcomatous differentiation: A rare case report. Medicine (Baltimore). 2023 Sep 29;102(39):e35465. doi: 10.1097/MD.0000000000035465. PMID: 37773783; PMCID: PMC10545380.

The potential role of the extracellular matrix in the activity of trabectedin in UPS and L-sarcoma: evidences from a patient-derived primary culture case series in tridimensional and zebrafish models. J Exp Clin Cancer Res. 2021 May 11;40(1):165. doi: 10.1186/s13046-021-01963-1. PMID: 33975637; PMCID: PMC8111914.

Paravertebral Well-Differentiated Liposarcoma with Low-Grade Osteosarcomatous Component: Case Report with 11-Year Follow-Up, Radiological, Pathological, and Genetic Data, and Literature Review. Case Rep Pathol. 2017;2017:2346316. doi: 10.1155/2017/2346316. Epub 2017 Mar 9. PMID: 28377828; PMCID: PMC5362705.

4.       Study limitations should be included

Author Response

Dear Reviewer,

Thank you very much for your valuable remarks and constructive comments. We have addressed them point by point. Additionally, in the revised version of the manuscript, changed or added text sections have been marked in yellow.

We are submitting a revised manuscript entitled "A rare case of dedifferentiated liposarcoma with osteosarcomatous differentiation- diagnostic and therapeutic challenges" and we hope that the revised version will be found satisfactory for publication in Diseases.

Kind regards,

Patrycja Sosnowska-Sienkiewicz

The authors Sosnowska-Sienkiewicz and colleagues report an uncommon presentation of dedifferentiated liposarcoma with osteosarcomatous differentiation.

  1. The manuscript is interesting and well described.

Re: Thank you sincerely for this comment. I am glad that the manuscript meets the conditions of a valuable publication.

  1. The manuscript would benefit from the followings:

  1. Latest WHO should be referenced: WHO Classification of Tumours. In Soft Tissue and Bone, 5th ed.; IARC Press: Lyon, France, 2020; Volume 3, p. 368, ISBN 978-92-832-4502-5.

Re: Thank you very much for this comment. This citation was added.

  1. Diagnostic images of MDM2 amplification should be reported (i.e. FISH, PCR etc)

Re: Thank you for this comment. Our patient's medical record contains the following comment:

"Amplification of the MDM2 gene was detected in the FISH test (result number LDF-273/23) - Genetic testing confirms diagnosis of dedifferentiated liposarcoma".

Unfortunately, other than this comment, we do not have access to the imaging results in the patient's medical record.

  1. The authors should include some reference from proper discussion

 Synchronous occurrence of small cell lung cancer and primary rectal dedifferentiated liposarcoma with osteosarcomatous differentiation: A rare case report. Medicine (Baltimore). 2023 Sep 29;102(39):e35465. doi: 10.1097/MD.0000000000035465. PMID: 37773783; PMCID: PMC10545380.

The potential role of the extracellular matrix in the activity of trabectedin in UPS and L-sarcoma: evidences from a patient-derived primary culture case series in tridimensional and zebrafish models. J Exp Clin Cancer Res. 2021 May 11;40(1):165. doi: 10.1186/s13046-021-01963-1. PMID: 33975637; PMCID: PMC8111914.

 Paravertebral Well-Differentiated Liposarcoma with Low-Grade Osteosarcomatous Component: Case Report with 11-Year Follow-Up, Radiological, Pathological, and Genetic Data, and Literature Review. Case Rep Pathol. 2017;2017:2346316. doi: 10.1155/2017/2346316. Epub 2017 Mar 9. PMID: 28377828; PMCID: PMC5362705.

Re: Thank you very much for this comment. All these citations were added to the text. The discussion was completed. Other additional citations marked in yellow were added to the manuscript.

  1. Study limitations should be included

Re: Thank you very much for this comment. The manuscript was significantly modified and completed.

The entire publication was also rechecked and revised by a native speaker.

Thank you again for all your comments.

Kind regards,

Patrycja Sosnowska-Sienkiewicz

Reviewer 3 Report

Comments and Suggestions for Authors

This case report describes a patient with retroperitoneal dedifferentiated liposarcoma with osteosarcomatous differentiation.  The case report may be of interest to the readership as heterologous osteosarcoma in dedifferentiated liposarcoma occurs occasionally.  An important reference to include is a paper published in 2018 by Yamashita, et al (Histopathology 2018 Apr (5), 729-38).  This paper was also referenced in the 2020 WHO Classification of Tumours 5th Ed. Soft Tissue and Bone Tumours.  In order to improve the case report, I have the following suggestions:

1) The English language needs considerable improvement.

2) I'm not sure what an ambiguous exit point represents (referring to the radiologic imaging).  

3) The photomicrographs should have a scale bar.  In Fig. 5, there is osteoblastic rimming, but cytologic atypia is not obvious and the picture is out of focus.  I would obtain low power and high power images.  Fig. 6 is too dark.  Since the tumor capsule was mentioned several times in the paper, I would include a photomicrograph of that, also.

Comments on the Quality of English Language

The English language in the manuscript can be improved.

In the Abstract, MDM2 should be italicized and the word "gene" used (lines 20, 26).

In line 33, tumor can be removed, since liposarcoma is a tumor.

In lines 53 and 54: Gene is missing the last e.  "Dedifferentiated components" instead of differentiated parts is suggested.

Line 58 angiosarcoma (maintain consistency).

Line 73 suggest "Diagnostics included"

Line 103 ? polycyclic

Line 111 I'm not sure what is meant by "around fields."

Line 114 FNCLCC should be written out first and the pathologic stage requires a citation.  

Line 168 could be stated better as: Amplification of the MDM2 gene...

Line 179 Positions?

Lines 185- 186 Sarcoma is malignant.  The sentence is awkward.

Line 199 Standard protocols exist for some neoplasms, even sarcomas.  Retroperitoneal sarcoma is not a specific diagnosis.

Line 206 I'm not sure what is meant by some histopathological findings, please clarify.

Author Response

Dear Reviewer,

Thank you very much for your valuable remarks and constructive comments. We have addressed them point by point. Additionally, in the revised version of the manuscript, changed or added text sections have been marked in yellow.

We are submitting a revised manuscript entitled "A rare case of dedifferentiated liposarcoma with osteosarcomatous differentiation- diagnostic and therapeutic challenges" and we hope that the revised version will be found satisfactory for publication in Diseases.

Kind regards,

Patrycja Sosnowska-Sienkiewicz

This case report describes a patient with retroperitoneal dedifferentiated liposarcoma with osteosarcomatous differentiation.  The case report may be of interest to the readership as heterologous osteosarcoma in dedifferentiated liposarcoma occurs occasionally.  An important reference to include is a paper published in 2018 by Yamashita, et al (Histopathology 2018 Apr (5), 729-38).  This paper was also referenced in the 2020 WHO Classification of Tumours 5th Ed. Soft Tissue and Bone Tumours.  In order to improve the case report, I have the following suggestions:

Re: Thank you very much for this comment. These two citations were completed. The content of the manuscript was changed. Other additional citations marked in yellow were added to the manuscript.

  • The English language needs considerable improvement.

Re: Thank you very much for this comment. The entire manuscript was linguistically revised.

  • I'm not sure what an ambiguous exit point represents (referring to the radiologic imaging).

Re:  According to the radiological evaluation, it was not possible to clearly determine the origin of the tumor based on the imaging studies performed.

  • The photomicrographs should have a scale bar. In Fig. 5, there is osteoblastic rimming, but cytologic atypia is not obvious and the picture is out of focus.  I would obtain low power and high power images.  6 is too dark.  Since the tumor capsule was mentioned several times in the paper, I would include a photomicrograph of that, also.

Re: Thank you for your comments. The figures 4, 5 and 6 were modified and completed according to advice. I consulted with our histopathologist about the capsule and its microphotograph. After discussion and the need to choose the best of the figures (limit). We decided to include the microphotograph of the capsule as a supplement (microphotograph of the capsule.jpg). 

Comments on the Quality of English Language

The English language in the manuscript can be improved.

In the Abstract, MDM2 should be italicized and the word "gene" used (lines 20, 26).

Re: Thank you very much. It was corrected.

In line 33, tumor can be removed, since liposarcoma is a tumor.

Re: Thank you very much. It was corrected.

In lines 53 and 54: Gene is missing the last e.  "Dedifferentiated components" instead of differentiated parts is suggested.

Re: Thank you very much. These were corrected.

Line 58 angiosarcoma (maintain consistency).

Re: Thank you very much. It was corrected.

Line 73 suggest "Diagnostics included"

Re: Thank you very much. It was corrected.

Line 103 ? polycyclic

Re: Thank you very much. It was corrected.

Line 111 I'm not sure what is meant by "around fields."

Re: Thank you very much. It was corrected.

Line 114 FNCLCC should be written out first and the pathologic stage requires a citation.

Re: Thank you for your comment. This is how our pathologists presented the final diagnosis in our documentation. Based on other publications, I changed this part and added the citation. Hope this is correct.

Line 168 could be stated better as: Amplification of the MDM2 gene...

Re: Thank you very much. It was corrected.

Line 179 Positions?

Re: Thank you very much. I’m very sorry. The modification was accidental by an automatic spell check of  Word. It was corrected.

Lines 185- 186 Sarcoma is malignant.  The sentence is awkward.

Re: Thank you very much. It was corrected.

Line 199 Standard protocols exist for some neoplasms, even sarcomas.  Retroperitoneal sarcoma is not a specific diagnosis.

Re: Thank you very much for this comment. Yes, of course there is no standard systemic adjuvant treatment for retroperitoneal sarcoma because it is not a specific histopathological diagnosis. It was corrected.

Line 206 I'm not sure what is meant by some histopathological findings, please clarify.

Re: Thank you very much for this comment. The appropriate text has been added to the manuscript.

The entire publication was also rechecked and revised by a native speaker.

Thank you again for all your comments.

Kind regards,

Patrycja Sosnowska-Sienkiewicz